# Efficacy of TurmiZn, a Metallic Complex of Curcuminoids-Tetrahydrocurcumin and Zinc on Bioavailability, Antioxidant, and Cytokine Modulation Capability

**DOI:** 10.3390/molecules28041664

**Published:** 2023-02-09

**Authors:** Dan DuBourdieu, Jamil Talukder, Ajay Srivastava, Rajiv Lall, Shital Panchal, Charmy Kothari, Ramesh C. Gupta

**Affiliations:** 1Probiotic Smart LLC, Menomonie, WI 54751, USA; 2Department of Pharmacology, Institute of Pharmacy, Nirma University, Ahmedabad 382481, India; 3Toxicology Department, Murray State University, Hopkinsville, KY 42240, USA

**Keywords:** TurmiZn, curcumin, tetrahydrocurcumin, zinc, bioavailability, antioxidant, cytokines

## Abstract

Complexes of curcumin with metals have shown much-improved stability, solubility, antioxidant capability, and efficacy when compared to curcumin. The present research investigates the relative bioavailability, antioxidant, and ability to inhibit inflammatory cytokine production of a curcuminoid metal chelation complex of tetrahydrocurcumin-zinc-curcuminoid termed TurmiZn. In vitro uptake assay using pig intestinal epithelial cells showed that TurmiZn has an ~3-fold increase (*p* ≤ 0.01) in uptake compared to curcumin and a ~2-fold increase (*p* ≤ 0.01) over tetrahydrocurcumin (THC). In a chicken model, an oral 1-g dose of TurmiZn showed a ~2.5-fold increase of a specific metabolite peak compared to curcumin (*p* = 0.004) and a ~3-fold increase compared to THC (*p* = 0.001). Oral doses (5 g/Kg) of TurmiZn in rats also showed the presence of curcumin and THC metabolites in plasma, indicating bioavailability across cell membranes in animals. Determination of the antioxidant activity by a 2,2-diphenyl-1-picryl-hydrazyl-hydrate (DPPH) radical scavenging assay indicated that TurmiZn was about 13x better (*p* ≤ 0.0001) than curcumin and about 4X better (*p* ≤ 0.0001) than THC, in reducing free radicals. In vitro experiments further showed significant (*p* ≤ 0.01) reductions of lipopolysaccharide (LPS)-induced proinflammatory cytokines such as interleukin (IL) IL-6, IL-8, IL-15, IL-18, and tumor necrosis factor (TNF)-alpha, while showing a significant (*p* ≤ 0.01) increase of granulocyte-macrophage colony-stimulating factor (GM-CSF) in dog kidney cells. In vivo cytokine modulations were also observed when TurmiZn was fed for 6 weeks to newborn chickens. TurmiZn reduced IL-1 and IL-6, but significantly reduced (*p* ≤ 0.01) IL-10 levels while there was a concurrent significant (*p* = 0.02) increase in interferon gamma compared to controls. Overall, these results indicate that TurmiZn has better bioavailability and antioxidant capability than curcumin or THC and has the ability to significantly modulate cytokine levels. Thus, TurmiZn could be an excellent candidate for a novel ingredient that can be incorporated into food and supplements to help overall health during the aging process.

## 1. Introduction

The overall health of an individual is supported by many factors, including nutrients that help the immune system. However, oxidative stress on the body is an underlying commonality for many disease conditions along with inflammation as aging occurs. Nutraceuticals containing curcuminoids and tetrahydrocurcumin (THC) have been used for both their capacity as antioxidants and their anti-inflammatory activity. Curcumin (1,7-bis(4-hydroxy-3-methoxyphenyl)-1,6-heptadiene-3,5-dione) is the main natural polyphenol found in turmeric (*Curcuma longa*) [1]. Turmeric has been used in Asian countries as a flavoring agent in various foods and as a traditional medicine against various health issues [2]. Turmeric and its constituents such as curcumin have numerous biological effects including antioxidant, anti-inflammatory, antiviral, antimicrobial, antifungal, anti-allergic, anticancer, antidiabetic, cardioprotective, nephroprotective, and neuroprotective properties [1]. In addition, curcumin can also affect the immune system [3] and cytokine levels. Curcumin’s effects include boosting certain cytokines such as beta interferon levels and reducing oxidative stress during influenza [4]. Curcumin metabolizes into various compounds including tetrahydrocurcumin. Curiously, THC is believed to have higher antioxidant capabilities than curcumin [5] but lower anti-inflammatory capabilities. THC is also known to affect cytokine levels [6].

Despite these multiple beneficial health properties, curcumin has some pharmacological issues. Some issues that hinder curcumin from being developed into a practical pharmaceutical candidate are its hydrophobic nature, water insolubility, poor bioavailability, rapid metabolism, and systemic elimination [7]. Several methods have been employed to increase the stability and bioavailability of curcumin, including binding curcumin to lipids [8,9], as well as creating phospholipid complexes of curcumin, albumin, and other molecules. It also has been noted that THC has better water solubility and antioxidant capability than curcumin [6] and is potentially a better candidate for pharmaceutical development for certain biological aspects. Efforts to improve the antioxidant capabilities of molecules and agents are known in the literature [10,11,12]. A new approach to improve curcumin’s antioxidant capability would be to complex curcumin to THC and take advantage of the individual capabilities of both molecules as a single entity. This approach has partially been explored [13] and has the potential to improve curcumin’s overall health benefits. Curcumin reacts with metals, such as zinc, through its β-diketone moiety to generate metal-curcumin complexes. It has been shown [14] that divalent zinc complexes with curcumin increase the stability of curcumin and increase its biological effects, under certain conditions, such as a significant reduction in colonic damage using curcumin-Zn complexes in experimentally induced ulcerative colitis. Zinc is now known to also chelate THC through its β-diketone moiety to form complexes [15]. The ability of zinc to chelate both β-diketone moieties of THC and curcumin simultaneously was used to create the novel TurmiZn complex [16]. Zinc is a trace mineral that has been shown [17] to proliferate immune cells via regulating cell division and thus exert immune-boosting activity. Curcumin and zinc have a history of antiviral activity in both preclinical and clinical studies [18]. Since curcumin has limited pharmacological effects, an approach to improve the efficacy of curcumin by utilizing zinc to chelate simultaneously with both curcumin and THC has been employed in this current research. This approach may prove to take advantage of the beneficial health effects of curcumin or curcumin-zinc complexes [19] but also, when complexed with THC via zinc in one chelation complex, to help modulate antioxidant and inflammation levels [20] by modulating the cytokine responses [21,22,23] in times of stress.

The use of curcuminoid-metal complexes and THC-metal complexes has only partially been explored. The novel TurmiZn complex of curcuminoids to THC via zinc chelation for improved bioavailability, antioxidant capability, and biological effects for cytokine modulation are reported in these current studies. It is hypothesized that TurmiZn will have greater bioavailability and antioxidant capability than curcumin due to the presence of zinc and tetrahydrocurcumin in the complex. These actions may then lead to the beneficial modulation of cytokine levels in cells that ultimately affect the body. 

## 2. Results

### 2.1. TurmiZn In Vitro Bioavailability

In the Transwell uptake experiments, absorbance was measured at both 220 nm and 425 nm to take into account the observed maximum absorbance of curcuminoids at 425 nm and the observed maximum absorbance of THC at ~220 nm. Both curcumin and THC have some absorbance in the 280 nm range. Zinc has no detectable interfering absorbance > 200 nm in this system. As the TurmiZn complex contains curcuminoids, zinc, and THC, absorbances at 425 nm predominantly reflect curcuminoids found in the complexes. Absorbances in the TurmiZn complex at 220 nm reflect mostly THC found in the complexes. Absorbances at 280 nm reflect a combination of tetrahydrocurcumin and curcuminoids. Figure 1a shows the percent net uptake of curcuminoids at 425 nm, the percent net uptake of THC at 220 nm, and the percent net uptake of the TurmiZn complex measured at both 220 nm and 425 nm and averaged together, using six replicates per group. The data of Figure 1a show an average 13% net uptake for curcuminoids, an average 17% net uptake for THC, and an average 43% net uptake of the TurmiZn complex in this system. These results indicate that TurmiZn complexes have a significantly elevated % net uptake across IPEC-J2 cell membranes as compared to either THC or curcuminoids. Between TurmiZn group and the curcumin group, *p* ≤ 0.01, while between the TurmiZn group and the THC group, *p* = 0.01. This represents a 3-fold increase in the net uptake of TurmiZn compared to curcuminoids and a 2-fold increase compared to THC.

The bioavailability of samples was also analyzed by using the HPLC method. The HPLC analysis performed at 280 nm of the Transwell upper and lower wells (Figure 1b) indicates that THC, curcuminoids, and the TurmiZn complex have the same basic respective HPLC profiles between the upper and bottom wells. This is consistent with these analytes, including TurmiZn, not being altered to any great extent by the actual passage through the intestinal cells used in this experiment. The HPLC analysis shows differences in peaks between the curcuminoids and THC while the TurmiZn complexes show chromatography profile similarities for both curcuminoids and THC. 

### 2.2. TurmiZn In Vivo Avian Bioavailability

A series of experiments were carried out in chickens to examine the uptake of TurmiZn in plasma as compared to curcuminoids and THC. The analytes were fed to three chickens and blood samples were taken at 2, 4, and 24 h to be processed for HPLC analysis at 425 nm as described in the methods. The chromatograms in Figure 2A–C indicate that HPLC peaks from the analytes are found in the plasma of chickens at least after 2 h of feeding, consistent with metabolites of the analytes being present after oral administration. Changes in certain peak heights and areas occurred over 24 h, consistent with the metabolism of each analyte.

The HPLC chromatograms of the TurmiZn complex (Figure 2C) show the appearance of a unique HPLC peak at the 1.4 min retention time in the 2 h plasma of chickens when fed a 1 g TurmiZn complex. That peak is reduced after 4 h and was undetectable after 24 h. A new HPLC peak at 15.5 min retention time in TurmiZn appears at 24 h. These peaks are presumed to be metabolites of the TurmiZn complex. The TurmiZn complex has major HPLC peaks that are also present in curcumin and THC profiles as seen at ~1.8 min and ~1.97 min. There are relative peak height differences between the curcuminoids and THC peaks whereby the 1.8 min peaks in curcuminoids are smaller than the 1.97 min peaks, with the reverse holding true for the THC equivalent peaks. The TurmiZn peaks at 1.8 and 1.97 min are similar in relative height to the THC peaks, but appear to metabolize at a different rate over time. The TurmiZn peak heights (Figure 2C) are all relatively higher than the curcuminoids and THC peaks. The height of the retention of the 1.8 min peak of the TurmiZn complex at 2 h in the chicken blood is about 2.5 times higher than the equivalent peak in curcuminoids (*p* = 0.004) or about 3-fold higher than the THC group (*p* = 0.001) (Figure 3A). This finding is consistent with a higher bioavailability of TurmiZn in chickens than curcuminoids or THC. Furthermore, the data show that the TurmiZn complex is metabolized over time, as evidenced by the disappearance of HPLC peaks and the appearance of new peaks in the serum at 24 h, at ~15.5 min retention time. These results are consistent with increased bioavailability and increased resident time of the TurmiZn complex in blood not seen with curcumin alone or THC alone.

### 2.3. TurmiZn In Vivo Mammalian Bioavailability

A series of in vivo experiments were carried out in rats to examine the uptake of TurmiZn into plasma. Rats were fed TurmiZn at a dose of 5 g/Kg and blood was drawn at specified time intervals and processed for HPLC analysis to detect the presence of metabolites of TurmiZn in the plasma as described in the methods. LC analysis of plasma at 2 h after oral delivery of TurmiZn indicated a major peak at 10.55 min retention time. This peak was not detected at time 0 before oral consumption of TurmiZn but a gradual increase of this peak began to occur by 15 min and was measured up to 2 h (Figure 3B). These results are consistent with metabolites of the TurmiZn complex being found in rodent plasma after oral administration.

Mass spectrometry analysis of the rat plasma was carried out as described in the methods on the blood taken at 45 min and at 2 h. The MS analysis shows numerous ions from the TurmiZn uptake into plasma. The results indicated numerous peaks at 45 min (Figure 4A) and then fewer MS peaks at 2 h (Figure 4B) for TurmiZn given orally to rats at a 5 g/Kg dose. The larger-sized MS peaks such as at 719 m/z and 796 m/z have largely disappeared by 2 h. The 719 m/z ion is consistent with being identified as curcumin diglucuronide. These results are consistent with TurmiZn ultimately being metabolized into smaller molecules.

An experiment to determine the presence of metabolites directly in plasma when TurmiZn was injected in the rat IV was carried out. Blood samples were removed periodically for up to 60 min and processed for HPLC analysis. Figure 5A shows a rapid decrease of a TurmiZn HPLC peak area of retention time of 10.5 min which occurred 15 min post IV injection. This TurmiZn peak is essentially gone by 45 min and is consistent with metabolism occurring of at least this particular peak of TurmiZn in rats within 1 h. 

The oral uptake of TurmiZn at 5 g/Kg dose in rat blood was monitored over the course of 4 h to examine chromatogram profiles for changes in peak appearances. Figure 5B shows a typical chromatogram profile of metabolite peaks found in rat plasma.

Table 1 shows that the TurmiZn metabolite at 12.108 min HPLC retention time has been decreased by 4 h while the peak at 3.344 min HPLC retention time has continued to increase over 4 h. The appearance of a TurmiZn metabolite at 2.375 min HPLC retention time not initially found at 1 h is found by 3 h. These data are consistent with the concept that when TurmiZn is fed to rats it is metabolized over the course of time into different metabolites.

### 2.4. Antioxidant Capabilities 

The results in Figure 6 of the DPPH antioxidant study show TurmiZn, curcuminoids, and THC exhibited antioxidant activity by scavenging free radicals for three replicates per group capability. The TurmiZn complex showed considerably higher antioxidant activity at all doses. TurmiZn had a 4-fold to 13-fold higher antioxidant capability as compared to the THC or curcuminoids respectively at the 0.125 mg/mL level. At the 0.125 mg/mL level, the *p*-value between the TurmiZn group and curcumin group is <0.0001 and the *p*-value between the TurmiZn group and THC group is <0.0001. The curcuminoids alone had the lowest antioxidant activity among them while THC exerted higher antioxidant activity than curcumin but less than TurmiZn. 

### 2.5. In-Culture Cytokine Production

In-culture cytokine production by lipopolysaccharide stimulation occurs in MDCK cells. Preincubation of MDCK cells with TurmiZn for 2 h prior to adding LPS significantly (*p* ≤ 0.01) reduces the release of the cytokines analyzed in Figure 7A–H in a dose-dependent manner. TurmiZn reduces LPS-stimulated IL-2 and IL8 in MDCK cells in a dose-dependent manner (Figure 7A,B). TurmiZn reduces LPS-stimulated IL-6 in MDCK cells in a dose-dependent manner, as seen in Figure 7C. TurmiZn eliminated LPS-stimulated IL-7 in MDCK cells, as seen in Figure 7D. TurmiZn reduces LPS-stimulated IL-15 and IL-18 in MDCK cells in a dose-dependent manner (Figure 7E,F). TurmiZn reduces LPS-stimulated MCP-1 levels, TNF alpha, and KC-like proteins in MDCK cells in a dose-dependent manner (Figure 7G–I).

While TurmiZn decreased, LPS stimulated the release of various proinflammatory cytokines, and TurmiZn also promoted the increased release of other cytokines. While LPS did not appear to affect GM-CSF levels, this cytokine was increased by TurmiZn in MDCK cells, as noted in Figure 7J.

### 2.6. TurmiZn In Vivo Cytokine Modulation

The release of cytokines by TurmiZn in animals was examined in chickens. TurmiZn was fed at 0, 0.5%, and 1% *w*/*w* rate in feed to newborn chickens daily for 6 weeks. At the end of the 6-week period, blood was drawn and processed for testing of certain cytokines by ELISA methodology. The results of cytokine levels in chickens fed the TurmiZn complex at the 1% rate for 6 weeks non-significantly reduced the amount of IL-1-beta (*p* = 0.07, Figure 8A) and IL-6 (*p* = 0.11, Figure 8B) compared to control chickens while significantly reducing IL-10 (*p* ≤ 0.01, Figure 8C) compared to control chickens. Orally fed TurmiZn in chickens concurrently significantly increased interferon-gamma levels (*p* = 0.02, Figure 8D) compared to control chickens.

## 3. Discussion

A fundamental aspect of several health issues is related to oxidative stress and inflammation [24]. It is believed that curcumin can help certain health issues through a variety of modes of action including its antioxidant capability and anti-inflammatory capability despite curcumin’s low bioavailability. As such, various methods have been used to modify the basic curcumin molecule to improve its bioavailability and antioxidant capability, and thus allow for greater health benefits. For example, free curcumin can improve its DPPH antioxidant activity when complexed to zinc anywhere from about 19% [25] to about 3-fold [26]. The mode of action of this increased antioxidant capacity of curcumin-zinc complexes appears to be related to stabilizing the structure of curcumin by the zinc [13]. This can improve health conditions as studies [14] indicated how zinc-curcumin complexes reduced experimentally induced ulcerative colitis for severity and extent of colonic damage. The current studies on TurmiZn may also prove useful to researchers examining health situations where oxidative stress leads to inflammation via cytokine modulation [27].

The current research uses zinc to complex THC to curcuminoids via chelation bonding to both entities. This approach allows for the known efficacies of THC to be combined with the efficacies of curcumin and curcuminoids along with the health benefits and efficacies of zinc in a single metallic curcuminoid/THC complex (TurmiZn). This metallic complex was subjected to its relative bioavailability and antioxidant capability compared to the THC and curcuminoids that were used to create the complex both under in vitro and in vivo conditions. The results of the current studies are consistent with the hypothesis that TurmiZn has greater bioavailability and antioxidant capability than free curcumin. In addition, the ability of TurmiZn was examined for its ability to affect cytokine levels under LPS-stimulated cytokines under in vitro conditions, and how basal levels can be affected under in vivo conditions as cytokine modulation is related to inflammation levels.

Curcumin has been found in previous studies to alleviate H_2_O_2_-induced oxidative stress and intestinal epithelial barrier injury along with mitochondrial damage in IPEC-J2 cells [28]. As such, we selected IPEC-J2 cells as a model for cellular uptake. The in vitro net uptake across cell membranes in Transwell culture for the intestinal cell line IPEC-J2 showed a statistically significant relative 3-fold increase of the TurmiZn compared to the curcuminoids, based on absorbance. The TurmiZn also showed a statistically significant relative 2-fold increase compared to the THC, as based on absorbance. These results are consistent with the TurmiZn being able to cross cell membranes more readily than the curcuminoids or THC presumably due to the presence of zinc in the complex. It is possible that the zinc complexation alters the structures of the curcuminoids and THC or polarity in a manner that allows for more efficient membrane crossing. It is also possible that a divalent metal transporter mechanism [29] is utilized by the metal curcuminoid complex to allow uptake. It is also possible the THC portion of the complex allows for more efficient uptake, possibly related to single bonds found in THC compared to double bonds found in curcumin. This would be consistent with published research indicating that THC is more water soluble and bioavailable than curcumin [5]. More studies are required to fully elucidate the modes of action to account for the increased bioavailability of TurmiZn.

IPEC-J2 cells used in this study are from intestinal porcine enterocytes isolated from the jejunum of a neonatal unsuckled piglet. The Transwell experiments indicated that as TurmiZn, curcuminoids, and THC crossed the IPEG-J2 cells, they did not have appreciably altered HPLC profiles. The results are consistent with the appreciable metabolism of TurmiZn complexes occurring after they have exited the intestinal cells. The in vivo data of TurmiZn, curcuminoids, and THC given to chickens and rats are consistent with the metabolism of these substances beginning within minutes. When identical amounts of TurmiZn, curcuminoids, or THC are fed to chickens, specific HPLC peaks are observed in plasma that are consistent with the TurmiZn complex having significantly greater bioavailability of potentially 3-fold, as measured by peak height and area, compared to the curcuminoids and THC that make up the TurmiZn. The peaks represent metabolites of TurmiZn, as certain peaks start to be reduced by 4 h and are undetectable by 24 h, while new peaks are found over time in the chromatograms.

Curcumin is known to exhibit very poor bioavailability. Numerous previous studies show very low, or even undetectable, concentrations in blood and extraintestinal tissue [7]. Some of the reasons are related to its poor absorption, rapid metabolism, chemical instability, and rapid systemic elimination. Orally delivered TurmiZn, on the other hand, can be detected in the blood. The current data indicate that TurmiZn-fed rats have components of TurmiZn continue to show up in plasma in increasing amounts for up to 2 h and then start to be metabolized, as based on HPLC and MS data. As based on direct IV injection of TurmiZn, once the TurmiZn and metabolites are present, they rapidly disappear within 15 min, presumably by further metabolism. It is possible that orally administered TurmiZn might have a reservoir pool that is slowly released from the GI tract into the blood to be quickly metabolized.

The metabolism of TurmiZn in rats and chickens may appear to be somewhat different in terms of exactly how the process works in regard to speed and details of the metabolites, although similar HPLC profiles appear to be found at 4 h post oral administration. However, it does appear that when TurmiZn is orally fed to animals such as chickens, it does cross intestinal cell membranes more efficiently than curcuminoids or THC alone and metabolites show up in plasma and ultimately disappear. What efficacy TurmiZn and its metabolites have in animals relative to curcuminoids and THC may in part be related to TurmiZn’s higher antioxidant capability. Curcumin bound to zinc previously has been shown to have a 3-fold improvement in antioxidant capability compared to free curcumin (30). In contrast to those studies, the current TurmiZn data show a considerably significant improvement in antioxidant activity. TurmiZn has up to a significantly 13-fold higher free radical reducing capability compared to free curcuminoids, depending on the relative dose. This ability may be related to the presence of zinc in the metallic complex of TurmiZn to help stabilize the physical geometry of the overall complex when free radicals bind to the phenolic structures on the complex. The THC component of TurmiZn may also be contributing to this antioxidant improvement. Further research is required to fully elucidate how TurmiZn improves antioxidant capability over free curcumin or curcumin-zinc complexes.

It is believed that curcumin has the potential for lowering inflammation. A meta-analysis study [27] in patients with an inflammatory background indicated that curcumin significantly reduces proinflammatory cytokines IL-1 and TNF-α, while IL-6 was reduced marginally in a non-significant manner while there were increases in IL-8 levels. The current studies show that TurmiZn can modulate cytokine releases under both in vitro and in vivo conditions. Curiously, while TurmiZn in the current studies significantly reduces IL-6, it also significantly reduces IL-8 levels, in contrast to previously reported meta-studies on just curcumin. Cytokines are small proteins that have a complex regulatory influence on inflammation and immunity, such as during infections. To mimic an infection, LPS present in the external wall of Gram-negative bacteria and largely responsible for their toxicity were used in conjunction with the MDCK cell line to stimulate cytokine release. These cells are widely used as epithelial cell models in studies ranging from vaccine production to viral infections. The in vitro data indicated that preincubation of the cells with TurmiZn helped reduce LPS-induced proinflammatory cytokine release in a dose-dependent manner. For example, TurmiZn significantly reduced LPS-stimulated IL-6 levels and eliminated the IL-7 response. IL-6 is associated with proinflammatory responses and responsible for many inflammatory diseases while increased levels of IL-7 cause cartilage matrix degradation that may result in joint destruction. Stimulated IL-2 and IL-8 levels were also significantly reduced by TurmiZn, as IL-2 can promote inflammatory responses while IL-8 plays a proinflammatory role as chemoattractant recruitment of neutrophils. LPS-stimulated proinflammatory cytokines IL-15 and IL-18 were significantly reduced by TurmiZn. Monocyte chemoattractant protein-1 (MCP-1) is one of the key chemokines that regulate migration and infiltration of monocytes/macrophages to produce inflammation and LPS-stimulated MCP-1 is significantly reduced by TurmiZn. Likewise, TNF binds to TNF receptors on immune cells initiating a cascade of cellular events that culminates in the release of inflammatory cytokines. LPS-stimulated TNF is significantly reduced by TurmiZn. TurmiZn significantly reduced Keratinocyte Chemotactic (KC)-like proteins levels induced by LPS. KC-like proteins are a predominant proinflammatory chemokine produced in glial cells upon infection with Theiler’s murine encephalomyelitis virus. In addition, TurmiZn can also increase levels of certain cytokines. For example, granulocyte-macrophage colony-stimulating factor (GM-CSF) that stimulates stem cells to produce granulocytes (neutrophils, eosinophils, and basophils) and monocytes was found to be increased by TurmiZn. An increased level of GM-CSF may be beneficial for immune system functions during times of stress.

The mode of action of these TurmiZn cytokine modulations may be through interactions with the transcription factor nuclear factor kB (NF-kB) and mitogen-activated protein kinase (MAPK) cascades that play fundamental roles among the intracellular signaling pathways involved in cytokine production, NF-kB is one of the most important inducible transcription factors in mammals and has been shown to play a pivotal role in the mammalian innate immune response [30] and chronic inflammatory conditions. MAPKs are stress-activated kinases that regulate many genes, such as ones involved in the inflammatory and immune response. NF-kB is considered the primary target of curcumin, and MAPKinase may also be affected. Curcumin inhibits the expression of LPS-induced inflammatory cytokines in macrophages using mechanisms that modulate the expression of the enzymes SOCS-1 and SOCS-3 and of p38 MAPK [31]. It may be possible that TurmiZn may also be affecting NF-kB and MAPKinase pathways to modulate cytokine levels. More research will be required to fully elucidate these mechanisms of action of TurmiZn under in vitro conditions.

TurmiZn also modulates cytokines in vivo. The current study indicates that newborn chickens fed on a daily basis for 6 weeks with TurmiZn will have altered levels of certain cytokines compared to control newborn chickens that did not consume TurmiZn. The control levels of proinflammatory IL-1beta and IL-6 were reduced, although not significantly, by TurmiZn. IL-1 beta was reduced by about 25% while IL-6 was reduced by about 50%. However, IL-10 was significantly reduced by TurmiZn and significant concurrent increases in beneficial interferon of about 7% occurred. These results indicate that in chickens fed TurmiZn, cytokine levels will be modulated.

## 4. Conclusions

The current results are consistent with the hypothesis that the TurmiZn complex has increased bioavailability and antioxidant capability relative to curcuminoids or THC. A potential mode of action that is consistent with the current data suggests that zinc helps to stabilize the curcumin-tetrahydrocurcumin chelation complex of TurmiZn. The stabilized complex can then cross membranes more readily than free curcumin or free THC, potentially due to divalent metal receptors in membranes, and/or with the help of THC. The complex then has better antioxidant capability than curcumin, potentially due to the complex being stabilized from a structural standpoint and the presence of THC in the complex. The complex can then affect the biological systems that modulate cytokines levels, as seen in the current in vitro and in vivo studies. As oxidative stress on the body is an underlying commonality for many disease conditions and inflammation and as aging occurs that also involves cytokines, these studies indicate that TurmiZn has excellent potential to be developed into an improved nutraceutical or into pharmaceutical preparations that have health benefits for both animals and humans.

## 5. Materials and Methods

### 5.1. Materials

Tetrahydrocurcumin (95% purity) and curcuminoid (95% purity) were purchased from Chemil Inc., Bothell WA. Methanol and anhydrous ZnCl_2_ (99.9% purity) were obtained from Sigma Chemicals, St. Louis, MO, USA. DPPH antioxidant assay kit was purchased from DOJINDO Laboratories, Kumamoto, Japan. Curcumin refers to curcuminoids in the remainder of this report unless designated otherwise.

### 5.2. TurmiZn Complexation Method

A ratio of 25:75 of THC:curcuminoid extract was created by weighing out relative amounts of THC and curcumin on a molar basis each and mixing them together to form a uniform powder (the ligand). A 2-mole amount (as based on curcumin) of ligand was added to a 1-mole amount of zinc chloride. Methanol was added to solvate the mixture and it was stirred at room temperature for 4 h. Complexation of the zinc to the curcuminoids and THC occurred during this time with an observed color change from orange to deep red. The solution containing complexed curcuminoids and THC to zinc was dried to a powder by roto-evaporation and termed TurmiZn. 

### 5.3. Bioavailability in Cells

Intestinal pig epithelial cells (IPEC) J2 cells were used and grown for 8 days to confluency in Corning™ 6 well Transwell™ multi-well plates with permeable polycarbonate membrane inserts, using Dulbecco’s Modified Eagle (DME) medium and 5% fetal calf serum [32,33]. The density of IPEC J2 cells was ∼2 × 10^6^ cells/well initially and was grown to confluence. The plates had a polyester membrane thickness of 10 μm, and 0.4 μm pore size. The cells were washed with Dulbecco phosphate-buffered saline. One milliliter of saline was present in the bottom wells at all times. Twenty milligrams of curcuminoids, THC, or TurmiZn complex were solubilized separately in 10 mL saline with 5.0% dimethylsulfoxide (DMSO), and 1 mL of the respective solutions was placed in the top of six replicate wells per treatment, and incubated for 2 h at 37 °C. Optical density (OD) at 220 nm and 425 nm was then measured in the top and bottom wells by ultraviolet/visible (UV/Vis) spectrophotometry (Figure 1A,B). The percent net average absorbance was calculated by dividing the bottom well absorbance by the total absorbances of the upper and bottom wells combined. High-performance liquid chromatography (HPLC) analysis was conducted on the upper and bottom wells at 280 nm on pooled samples (Figure 2A). HPLC analysis was carried out with a Waters 2965 HPLC coupled with a photodiode array (PDA) detector (model 2996) along with Empower software. Column: Phenomenex Luna C18 250 × 4.6 mm, 5 µM. Isocratic: 50:50 acetonitrile: 2% acetic acid in water. Flow rate 1 mL/min for 20 min. The extraction solvent for analytes was acetone. The column temperature was 25 °C. PDA wavelengths 425 nm and 220 nm were used. The mean values ± standard error are reported. 

### 5.4. Bioavailability in Animals

Animal trials were conducted to determine the bioavailability of the TurmiZn complex in rats and chickens. Rats are a more suitable mammalian model than mice to perform kinetic studies that require blood withdrawal for longer durations. Chickens are suitable models for bioavailability studies for researchers interested in avian production. Doses were selected based on the ability to readily monitor TurmiZn metabolites analytically in plasma and were not based on therapeutic levels. 

For the avian bioavailability trials, chickens (Rhode Island Red, 3 chickens/group) that were each 2 years old and weighing 5 pounds were selected, and 1 mL of blood was drawn from the wing vein for baseline measurements. The blood was allowed to clot and the plasma was removed by centrifugation for subsequent baseline analysis. The chickens were allowed to rest overnight. One gram of TurmiZn, one gram of curcumin, and one gram of THC were suspended in 5 mL of water and placed into respective syringes. The contents of each of the respective syringes were then administered orally ensuring that the contents were consumed. Blood was drawn at 2, 4, and 24 h from each bird and allowed to clot. The recovered serum was mixed with equal volumes of acetone via vortexing for 30 s and then centrifuged to remove proteins [34,35] in the pellet while recovering curcuminoids, THC, or elements of TurmiZn in the supernatant. The supernatants from each time point were analyzed by HPLC methodology as described for the presence of curcuminoids, THC, or TurmiZn at 425 nm.

Bioavailability study in rats was approved by IAEC (Institutional Animal Ethics Committee) of the Institute of Pharmacy, Nirma University (Ahmedabad, India) as per the CPCSEA (Committee for Control and Supervision of Experiments on Animals) guidelines, Department of Animal Husbandry and Dairying (DAHD), Ministry of Fisheries, Animal Husbandry and Dairying (MoFAH&D), Govt. of India. The protocol number was IP/PCOL/FAC/29/2021/32. The bioavailability was evaluated in adult healthy male Wistar rats with two different studies: oral administration and intravenous administration of TurmiZn complex as a single dose. For orally administered trials in mammals, baseline blood samples were taken and three animals were administered TurmiZn 5 g by mouth. Blood samples were then withdrawn at 15-minute intervals up to 60 min and then hourly for up to 4 h and allowed to clot. The recovered serum was mixed with equal volumes of acetone via vertexing for 30 s and then centrifuged to remove proteins in the pellet while recovering elements of TurmiZn in the supernatant. The samples were analyzed by HPLC and liquid chromatography-mass spectrometry/mass spectrometry (LC/MS/MS). In another study, intravenous (IV) injection of TurmiZn in a single healthy adult Wistar rat was used to obtain metabolic profiles of TurmiZn over time. This was based on a protocol [36] for the pharmacokinetics and tissue distribution of curcumin in mice. TurmiZn was injected (100 mg/Kg, IV) and blood was collected at 2, 5, 15, 30, 45, and 60 min and allowed to clot and extracted as previously described in acetone for ultra-fast liquid chromatography (UFLC), LC-MS/MS analysis. 

### 5.5. HPLC Analysis for Avian Oral Dosage Samples

HPLC analysis of chicken serum was carried out with a Waters 2965 HPLC coupled with a PDA detector (model 2996) along with Empower software. Column: Phenomenex Luna C18 250 × 4.6 mm, 5 µM. Isocratic: 50:50 acetonitrile: 2% acetic acid in water. Flow rate 1 mL/min for 20 min. The extraction solvent for analytes was acetone. The column temperature was 25 °C. PDA wavelengths 425 nm, 280 nm, and 220 nm were used.

### 5.6. LC-MS/MS Analysis for Mammalian Oral Dosage Samples

For LC-MS/MS of rat serum samples after oral administration, the HPLC (Shimadzu Corp, Japan) system consisted of an auto-sampler (Shimadzu SIL-20AC), column oven (CTO-20AC), quaternary gradient pump (LC-20ADvp), degasser unit (DGU-20A5R), and system controller (CBM-20A and SPD-M20A). PDA detection was used. The LC-MS (8030, Shimadzu Corp, Japan) consisted of ESI as an ion source in positive and negative mode, mass scan ranges 50–1000 Da, collision energy was 10, 20, and 30 eV, DL temperature was 250° C, nebulizer gas flow was 3.0 L/min, heat block temperature was 350 °C, and drying gas flow was 15.0 L/min. LabSolutions was used as the operating software. Separation was achieved using a Nucleosil C18 (250 mm × 4.6 mm, 5 μm) column with a mobile phase consisting of water with 2% acetic acid: acetonitrile (50:50, *v*/*v*) in an isocratic mode at 1.2 mL/min flow rate. The column temperature was 33 °C and the autosampler temperature was 15 °C. The injection volume was 20 μL. The run time was 20 min. The wavelengths were 280 nm and 425 nm.

### 5.7. LC-MS/MS Analysis for Mammalian IV Dosage Samples

For LC-MS/MS of rat serum samples after direct IV administration, the ultra-fast liquid chromatograph (UFLC) (Shimadzu Corp, Kyoto, Japan) system consisted of an autosampler, column oven, quaternary gradient pump, degasser unit, and system controller. The UFLC-MS (Shimadzu Corp, Kyoto, Japan) consisted of ESI as an ion source in positive and negative mode, mass scan ranges 50–1000 Da, collision energy was 25 eV, the delustering potential was 70 eV, entrance potential was 10 eV, and collision cell exit potential was 10 eV. DL temperature was 250 °C, nebulizer gas flow was 3.0 L/min, heat block temperature was 350 °C, and drying gas flow was 15.0 L/min. LabSolutions was used as the operating software. Separation was achieved using a Nucleosil C18 (100 mm × 3.1 mm, 2 μm) column with a mobile phase consisting of water with 2% acetic acid: acetonitrile (50:50, *v*/*v*) in an isocratic mode at 0.2 mL/min flow rate. The column temperature was 33 °C and the autosampler temperature was 15 °C. The injection volume was 10 μL and the run time was 15 min.

### 5.8. Cytokine Determination in-Culture Methods 

Mardin-Darby canine kidney (MDCK) cells were grown on 6-well plates with nested inserts (growth area 4.67 cm^2^, 0.4 µm pore size) with DME media containing 5% fetal calf serum (FCS) under 5% CO_2_ and 95% humidity (6 wells per group). Initially, ∼2 × 10^6^ cells were placed on top of the insert per well. Then, 5-days post-confluent cells were washed with phosphate-buffered saline. Lipopolysaccharide (LPS, 8 µg/mL) and different concentrations of TurmiZn were used in 3 mL of DME without FCS per well. TurmiZn concentrations were (mg/mL): 0.0, 0.125, 0.25, 0.5, 1.0, 2.0, and 4.0. Cells were pretreated with TurmiZn for 2 h then LPS was added in treated groups. Cells were then incubated for 24 h. Cytokines were measured using the Canine Cytokine Array/Chemokine Array 13-Plex (CD13) assay (www.evetechnologies.com (accessed on 7 February 2023)). The (-) indicates negative control while a (+) indicates positive control. Data were statistically analyzed using GraphPad Prism. Mean values ± standard error of the mean are reported.

### 5.9. Cytokine In Vivo Methods

Three groups each containing three one-day-old chicks (Cornish cross males) were placed in three identical pens of identical sizes (5 foot × 5 foot), with adequate light, heat, water, and free access to food. The first chicken group served as a control, the second group received 0.5% *w*/*w* of the TurmiZn complex mixed into the feed, and the third group received 1.0% TurmiZn mixed into the feed. The feed used throughout the trial was standard poultry feed mash to ensure uniform mixing. At the end of the 6-week period, blood samples were taken and allowed to clot into a serum. The serum was tested by commercial ELISA kits (Cusbio) specific for chicken interleukins (IL) IL-1beta, chicken IL-6, chicken IL-10, and chicken interferon-gamma using directions provided by the manufacturer. The data are reported as mean values ± standard error of the mean.

### 5.10. Antioxidant Methods

The antioxidant assay was carried out by the DPPH method. The DPPH (2,2-diphenyl-1-picryl-hydrazyl-hydrate) free radical method is an antioxidant assay, based on electron transfer that produces a violet solution in ethanol. This free radical, stable at room temperature, is reduced in the presence of an antioxidant molecule, giving rise to a yellow-colored product, diphenylpicryl hydrazine in ethanol solution. The DPPH assay is used to determine antioxidant activities by the mechanism in which antioxidants act to inhibit lipid oxidation, scavenging of DPPH radical, and therefore determine free radical scavenging capacity. Free radical scavenging abilities of curcumin, TurmiZn, and THC were tested by the DPPH radical DPPH antioxidant assay kit (Dojindo Molecular Technologies, Inc., Rockville, MD, USA). Briefly, for three replicates of each analyte, a solution of 10 mL DPPH in ethanol was prepared, and 100 μL of this solution was mixed with 20 μL of the curcuminoids, THC or TurmiZn complex solubilized in ethanol at different concentrations (0.125–0.50 mg/mL). The reaction mixture was mixed thoroughly and left in the dark at room temperature for 30 min. The absorbance of the mixture was measured spectrophotometrically at 517 nm. Trolox was used as a reference. The percentage DPPH radical scavenging activity was then calculated per manufacturing methodology and reported as mean values ± standard error of mean.

### 5.11. Statistical Analysis

Statistical analyses were performed using GraphPad Prism statistical software. The differences between the mean values of groups were analyzed with the use of the single-tail unpaired Student’s *t*-test. Differences were considered significant at *p* < 0.05. Unless otherwise indicated, the results are reported as mean values with their standard error of the mean.

## Figures and Tables

**Figure 1 molecules-28-01664-f001:**
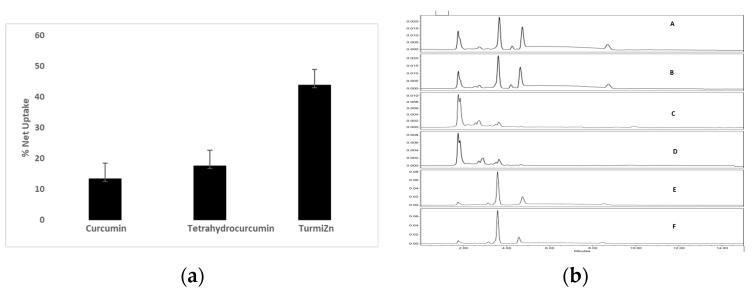
(**a**) Net uptake of curcumin, THC, and TurmiZn across IPEC-J2 cells. subfigure (**a**) shows the percent net uptake of curcuminoids at 425 nm, the percent net uptake of THC at 220 nm, and the percent net uptake of the TurmiZn complex measured at both 220 nm and 425 nm and averaged together, using six replicates per group. (**b**) Transwell HPLC chromatograms at 280 nm. subfigure (**b**) indicates that THC, curcuminoids, and the TurmiZn complex have the same basic respective HPLC profiles between the upper and bottom wells. A: TurmiZn top well; B: TurmiZn bottom well; C: curcumin top well; D: curcumin bottom well; E: THC top well; F: THC bottom well. Values are mean ± standard error of mean.

**Figure 2 molecules-28-01664-f002:**
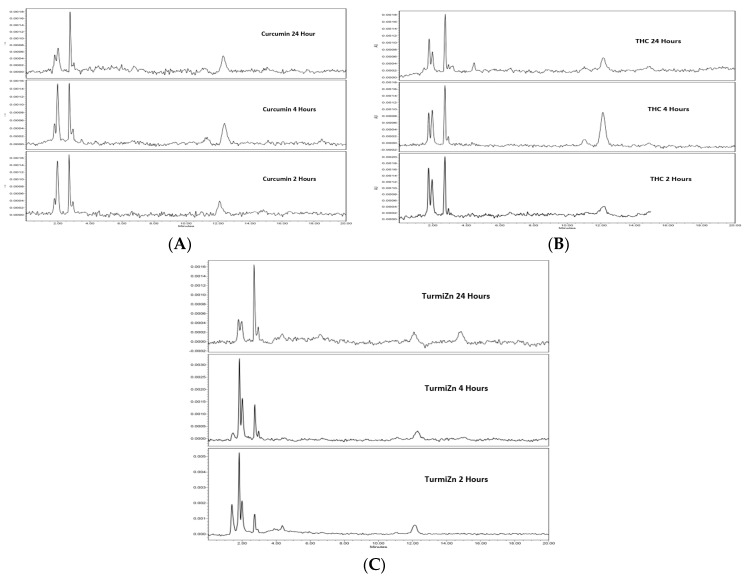
(**A**) HPLC chromatograms of curcuminoids in chicken plasma 2, 4, and 24 h post oral administration at 425 nm. (**B**) THC in chicken plasma at 2, 4, and 24 h post oral administration at 425 nm. (**C**) TurmiZn (THC-Zn-Curcumin) in chicken plasma at 2, 4, and 24 h post oral administration at 425 nm.

**Figure 3 molecules-28-01664-f003:**
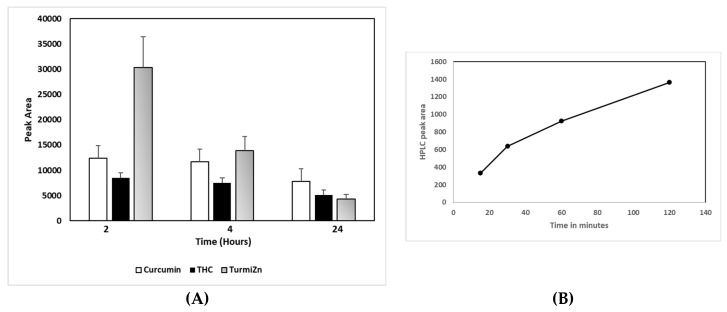
(**A**) TurmiZn HPLC peak area at 1.8 minutes retention time over 24 hours in chicken plasma. (**B**) Time course of oral TurmiZn in rat plasma of LC peak area at 10.55 min retention time. Values are mean ± standard error of the mean.

**Figure 4 molecules-28-01664-f004:**
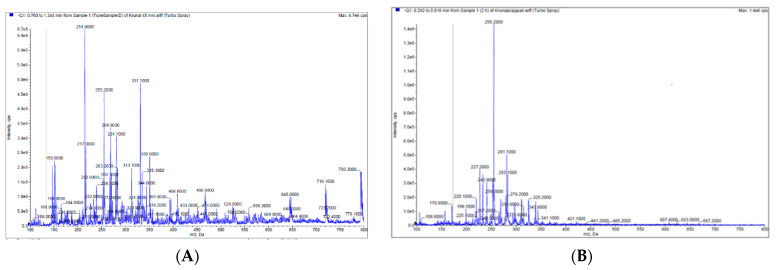
(**A**) MS of TurmiZn complex in rat plasma sample (45 min) of 5 g/Kg oral dose). (**B**) MS of TurmiZn in rat plasma sample (2 h) of 5 g/Kg oral dose).

**Figure 5 molecules-28-01664-f005:**
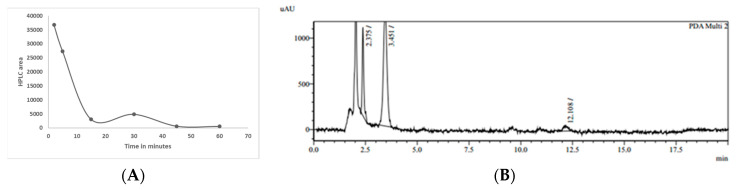
(**A**) Time course of TurmiZn after IV injection in rats. (**B**) Representative HPLC chromatogram of TurmiZn fed to rats (5 g/Kg dose) in plasma at 4 h at 425 nm.

**Figure 6 molecules-28-01664-f006:**
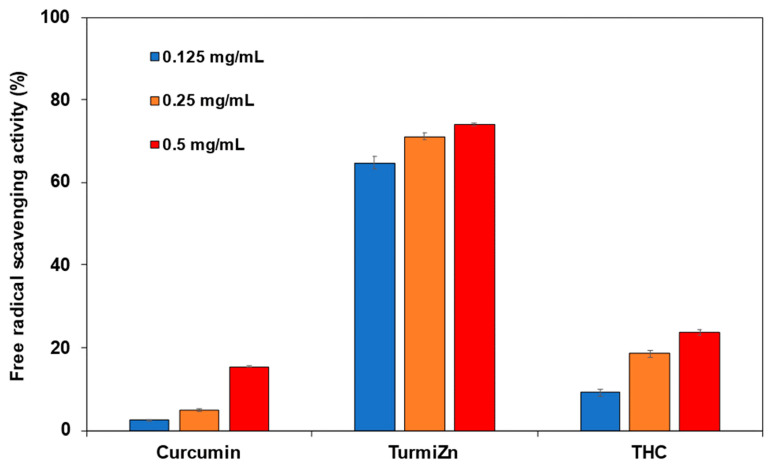
Antioxidant capability of TurmiZn, curcumin, and THC dose response in the DPPH assay. Values are mean ± standard error of mean.

**Figure 7 molecules-28-01664-f007:**
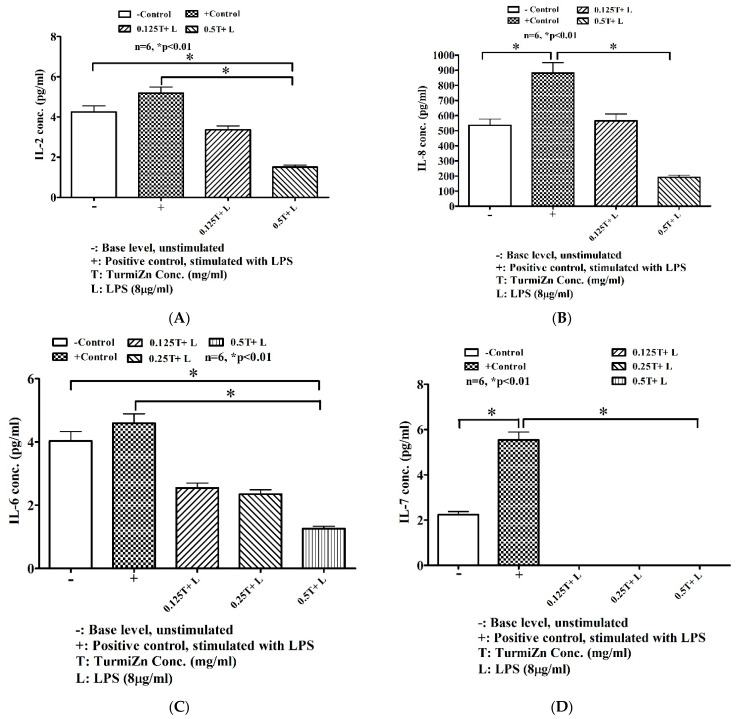
TurmiZn modulates LPS-induced cytokine production in MDCK cells in a dose-dependent manner. (**A**) IL-2 levels; (**B**) IL-8 levels, (**C**) IL-6 levels; (**D**) IL-7 levels (**E**) IL-15 levels; (**F**) IL-18 levels; (**G**) MCP-1 levels; (**H**) TNF levels; (**I**) KC-like protein levels; and (**J**) GMCSF levels. Values are mean ± standard error of the mean.

**Figure 8 molecules-28-01664-f008:**
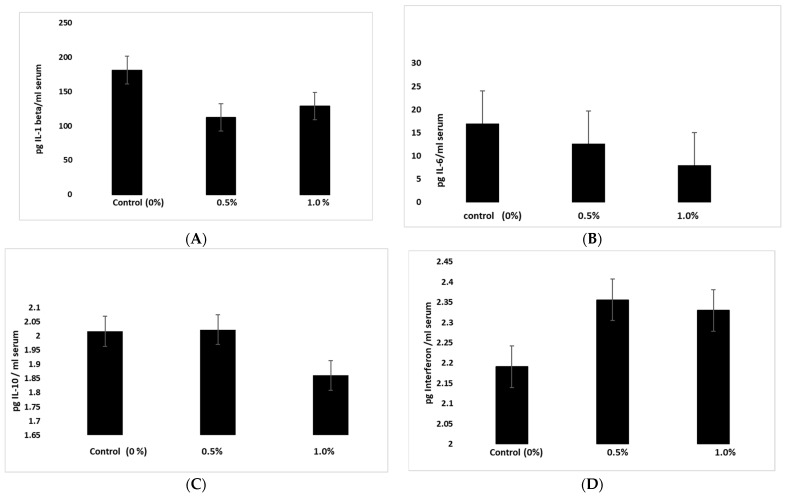
TurmiZn modulates levels of cytokines in broilers fed at 0 to 1% *w*/*w* inclusion rates for 6 weeks. (**A**) IL-1 beta levels; (**B**) IL-6 levels; (**C**) IL-10 levels; (**D**) Interferon levels. Values are mean ± standard error of mean.

**Table 1 molecules-28-01664-t001:** Peak areas of TurmiZn from Figure 5B in plasma from oral administration over 4 h.

Hour	Peak Area 2.375 Min	Peak Area 3.441 Min	Peak Area 12.108 Min
1	0	2160	1662
2	0	2484	4791
3	1158	4594	1672
4	5052	15,701	1037

## Data Availability

Data is contained within the article.

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
