# Peer review of "Efficacy of TurmiZn, a Metallic Complex of Curcuminoids-Tetrahydrocurcumin and Zinc on Bioavailability, Antioxidant, and Cytokine Modulation Capability"

_molecules, 2023, doi:10.3390/molecules28041664_

Round 1

Reviewer 1 Report

Recommendation letter

Dear Dr.

I have read the manuscript (molecules-2176285-peer-review-v1) under title of “Efficacy studies of TurmiZn, a metallic complex of curcuminoids-tetrahydrocurcumin and zinc, on bioavailability, antioxidant, and cytokine modulation capability” will considered for publication in Molecules, I have the following concerns about this manuscript.

1.  The article is very useful for researchers in this field, but it is need revisions before provide the final form of publishing.

2.  The written structure should be done upon standard, like space between word, standard units ……..etc, should be revised.

3.  Abbreviations should be written in complete form at the first time.

4.   

5.  Authors should improve the manuscript with the following recent literatures: https://doi.org/10.1016/j.btre.2022.e00768., https://doi.org/10.1016/j.btre.2022.e00753., https://doi.org/10.1016/j.btre.2019.e00377.

6.  Statistical analyses should be provided.

7.  The figures should be improved and cleared (it is appeared in lowest format).

8.  The figures denied the statistical analyses.

9.  The article needs some graphical figures to improve its quality.

Author Response

Dear Dr.

I have read the manuscript (molecules-2176285-peer-review-v1) under title of “Efficacy studies of TurmiZn, a metallic complex of curcuminoids-tetrahydrocurcumin and zinc, on bioavailability, antioxidant, and cytokine modulation capability” will considered for publication in Molecules, I have the following concerns about this manuscript.

  1. The article is very useful for researchers in this field, but it is need revisions before provide the final form of publishing. Reply: Extensive revisions have occurred throughout the manuscript.
  2. The written structure should be done upon standard, like space between word, standard units ……..etc, should be revised.  Reply: written structure now uses standard spacing and units.
  3. Abbreviations should be written in complete form at the first time. Reply: abbreviations are now explained in the text
  4.  
  5. Authors should improve the manuscript with the following recent literatures: https://doi.org/10.1016/j.btre.2022.e00768., Now cited. https://doi.org/10.1016/j.btre.2022.e00753.,  Now cited. https://doi.org/10.1016/j.btre.2019.e00377.  Now cited.
  6. Statistical analyses should be provided. Reply: Statistics are now provided in graphs and in text.
  7. The figures should be improved and cleared (it is appeared in lowest format). Reply: Figures are improved with higher resolutions.
  8. The figures denied the statistical analyses. Reply: Statistics of figures are now provided and discussed in the text.
  9. The article needs some graphical figures to improve its quality  Reply: Graphics of figures have higher resolutions now.

Reviewer 2 Report

The manuscript entitled “Efficacy studies of TurmiZn, a metallic complex of curcuminoids-tetrahydrocurcumin and zinc, on bioavailability, antioxidant, and cytokine modulation capability” by DuBourdieu et al. intends to evaluate the effects of complexes of curcumin – tetrahydrocurcumin-zinc-curcuminoid in bioavailability, antioxidant, and inhibition of inflammatory cytokine, using in vitro and in vivo studies. The manuscript is in the scope of Molecules. The paper is interested. However, there are major issues regarding the scientific design of the experiments that should be addressed:

In the Abstract, line 22 introduce IL before – interleukin.

In the introduction, the authors should revise and improve with references (such as in lines 38, 45, 54). The references should be formatted as described in authors’ guide.

The sentence of introduction (lines 61-63) should be rewrite and improved.

In the materials and methods sections, the use of abbreviations should be revised, words as DMSO, DME, OD, … please introduce before.

The section 2.3 should be sub-divided.

What is the cell density used in IPEC J2 cells?

For in vivo studies, how many animals were used? Please described better the methods used.

The “DMEM medium” described in line 174 is the same used in line 102? Please uniformize.

In line 194, the abbreviation should be introduced firstly.

Figure 2B presents low quality. The figures 2 have not yy axis. Please revise.

Figure 6 also has low quality. Please pay attention.

The discussion section should be improved with references, namely in lines 432-456. In this section, the authors should compare the results with other studies and after detail the conclusions.

Author Response

The manuscript entitled “Efficacy studies of TurmiZn, a metallic complex of curcuminoids-tetrahydrocurcumin and zinc, on bioavailability, antioxidant, and cytokine modulation capability” by DuBourdieu et al. intends to evaluate the effects of complexes of curcumin – tetrahydrocurcumin-zinc-curcuminoid in bioavailability, antioxidant, and inhibition of inflammatory cytokine, using in vitro and in vivo studies. The manuscript is in the scope of Molecules. The paper is interested. However, there are major issues regarding the scientific design of the experiments that should be addressed:

In the Abstract, line 22 introduce IL before – interleukin. Reply: This has been revised to define all abbreviations in the text.

In the introduction, the authors should revise and improve with references (such as in lines 38, 45, 54). The references should be formatted as described in authors’ guide. Reply:  Additional references have been added  at these basic locations and formatted as described in the guide.

The sentence of introduction (lines 61-63) should be rewrite and improved.  Reply: The introduction has been revised and improved at these lines.

In the materials and methods sections, the use of abbreviations should be revised, words as DMSO, DME, OD, … please introduce before.  Reply:  descriptions of abbreviations has occurred throughout the text in the revision.

The section 2.3 should be sub-divided. Reply:  The section has been subdivided.

What is the cell density used in IPEC J2 cells? Reply: The plating cell density is now stated and  plate areas.

For in vivo studies, how many animals were used? Please described better the methods used.  Reply: The number of animals for in vivo studies is now described in the text.  The methods are now described in a better manner in the text.

The “DMEM medium” described in line 174 is the same used in line 102?  Please uniformize.  Reply:  The media is now more clearly described in the text.

In line 194, the abbreviation should be introduced firstly. Reply: Word descriptions are now defined in the text prior to the abbreviations.

Figure 2B presents low quality. The figures 2 have not yy axis. Please revise.  Reply:  Figure 2B has been revised to have higher resolutions and  have  xy axis visible.

Figure 6 also has low quality. Please pay attention.  Reply:; Figure 6 has been revised to have higher resolution.  All figures now have higher resolution.

The discussion section should be improved with references, namely in lines 432-456. In this section, the authors should compare the results with other studies and after detail the conclusions.  Reply:  The discussion section has been revised and improved  to include references, to compare the current to prior data and expand the conclusions of the current data.

Reviewer 3 Report

Regarding MS entitled ‘’ Efficacy studies of TurmiZn, a metallic complex of curcuminoids-tetrahydrocurcumin and zinc, on bioavailability, antioxidant, and cytokine modulation capability’’

Title

Delete studies and comma after zinc

Abstract

Please add p-value for significant findings

Introduction

L82 please add the hypothesis

M&M

L109. Please add more information about HPLC

For animal, avian, and rat trials, the number of animals used and the number of sampling, as well as replication should be added. The length of the trial for animals should be added. Why did the authors choose this dose, and on which basis?

There is no information about the weight of animals and how these animals were selected.

L129. Add ref

L139. Add ref

The statistical analysis part is missing. Please add

Results

About figures, how was the obtained data analysed? And data in figures are presented as mean ± what?

Please add p value for significant findings.

About figure 7, the error bars (SEM or SD) are missing.

The discussion section is very poor and needs extensive revisions from the authors. The authors did not discuss their findings with earlier studies. There is not any cited ref!!

Discussion lacks mechanisms of action and conceptualization.

Author Response

Comments and Suggestions for Authors

Regarding MS entitled ‘’ Efficacy studies of TurmiZn, a metallic complex of curcuminoids-tetrahydrocurcumin and zinc, on bioavailability, antioxidant, and cytokine modulation capability’’

Title

Delete studies and comma after zinc  Reply: Title has been revised accordingly.

Abstract

Please add p-value for significant findings  Reply: p values have been added in the revision.

Introduction

L82 please add the hypothesis  Reply:  The hypothesis has been added in the revision.

M&M

L109. Please add more information about HPLC  Reply: More information has been added about the HPLC.

For animal, avian, and rat trials, the number of animals used and the number of sampling, as well as replication should be added. The length of the trial for animals should be added. Why did the authors choose this dose, and on which basis?  Reply:  The number of animals, trial lengths, and replicates of assays have been added in the revised text. The basis for doses is now described in the revised text.

There is no information about the weight of animals and how these animals were selected. Reply: Weights of animals are described and selected in the revised text

L129. Add ref Reference is now added in revised text

L139. Add ref  Reference is now added in the revised text.

The statistical analysis part is missing. Please add  Reply:  Statistical analysis has been added to the data in and the text.

Results

About figures, how was the obtained data analysed? And data in figures are presented as mean ± what?  Reply:  statistical analysis has been added and the data is now described as mean +/- standard error of mean in the revised text.

Please add p value for significant findings.  Reply: p values have been added for data for significant findings

About figure 7, the error bars (SEM or SD) are missing. Reply:  Figure 7 has error bars added in the revised graph.

The discussion section is very poor and needs extensive revisions from the authors. The authors did not discuss their findings with earlier studies. There is not any cited ref!!  Reply:  Extensive revision of the discussion has occurred in the revised text.  Earlier studies are discussed and compared to the current data in the revised text.

Discussion lacks mechanisms of action and conceptualization.  Reply: A mechanism for the mode of action is now discussed in the revised text of the discussion.

Round 2

Reviewer 1 Report

the manuscript is accepted in this form

Author Response

Thank you for your review.  No further changes.

Reviewer 2 Report

The authors modified and improved manuscript according to the comments. The manuscript should be accepted in present form. 

Author Response

Thank you for your review. No further changes added.

Reviewer 3 Report

Thank you for revisions. The statistical analysis should be presented in a separate heading under materials and methods section. I did not notice it in the revised version. 

Author Response

Comment: The statistical analysis should be presented in a separate heading under materials and methods section. I did not notice it in the revised version. 

reply:  Statistical analysis has now  been added in materials section 2.11 (in red font)

Thank you for your review.